# Effects of Citric Acid, Synbiotic, and Probiotic Supplementation Through Drinking Water on Growth Performance, Carcass Yield, and Blood Biochemistry of Broiler Chickens

**DOI:** 10.3390/ani15081168

**Published:** 2025-04-18

**Authors:** Shahadot Hossain, Biswajit Kumar Biswas, Subir Das, Faija Sadia Pory, Rabin Raut, Fatima Yeasmin, Sharif Uddin Khan, Prantho Malakar Dipta, Sabbir Alom Shuvo, Tahera Yeasmin, Raihanul Hoque

**Affiliations:** 1Department of Dairy and Poultry Science, Hajee Mohammad Danesh Science and Technology University, Dinajpur 5200, Bangladesh; shahadot43258@bau.edu.bd (S.H.); taherayb@yahoo.com (T.Y.); 2Department of Food and Animal Sciences, Tennessee State University, Nashville, TN 37209, USA; 3Department of Poultry Science, Auburn University, Auburn, AL 36849, USA; rzr0096@auburn.edu; 4College of Engineering & Science, Molecular Science and Nanotechnology, Louisiana Tech University, Ruston, LA 71272, USA; fatimayeasmin321@gmail.com; 5Department of Animal Science, Animal Molecular & Cellular Biology, University of Florida, Gainesville, FL 32611, USA; 6Department of Poultry Science, Center of Excellence for Poultry Science, University of Arkansas, Fayetteville, AR 72701, USA; 7College of Engineering & Science, Computational Analysis and Modeling, Louisiana Tech University, Ruston, LA 71272, USA; shovon961@gmail.com

**Keywords:** broilers, water additives, growth performance, meat yield, blood parameters

## Abstract

Poultry farming is experiencing rapid growth in Bangladesh, where the use of antibiotics as growth promoters is still common. However, with rising concerns about antibiotic resistance and changing market demands, there is an urgent need to identify effective non-antibiotic alternatives, especially for small- and medium-scale poultry farmers. This study was designed to investigate the effects of citric acid, synbiotics, and probiotics supplemented through drinking water on the growth, carcass yield, and blood metabolites of broiler chickens. The probiotic group achieved higher body weight, weight gain, and better feed efficiency compared to the other groups. Additionally, probiotics showed the best results regarding beneficial changes in blood parameters, including cholesterol and low-density lipoprotein levels. In conclusion, while citric acid and synbiotics also provide noticeable improvements, probiotics are the most effective alternative for enhancing broiler growth and overall performance in poultry farming.

## 1. Introduction

The poultry sector in Bangladesh is rapidly shifting from traditional backyard farming to commercial production systems, with an annual increase of approximately 15% in commercial poultry farms to meet the protein demands of the growing population [1]. In Bangladesh, most commercial poultry farms use antimicrobials during production and for prophylactic purposes [2]. However, rising consumer demand for antibiotic-free meat and concerns over antimicrobial resistance (AMR) from regulatory bodies have led to stricter policies. In 2010, the Bangladesh government passed the “Bangladesh Fish Feed and Animal Feed Act 2010”, prohibiting the inclusion of antibiotics, growth hormones, steroids, and insecticides in animal feed during manufacturing [3]. While antimicrobial restrictions aim to mitigate AMR, they also present significant challenges if not properly addressed. Seasonal variations and ignorance of other factors such as biosecurity, litter management, water sanitation, and stocking density create favorable conditions for microbial growth, compromising bird health [4,5]. Therefore, identifying cost-effective alternatives that maintain broiler health while promoting faster growth is essential, especially considering the cost management constraints in poultry production.

Several non-antibiotic growth promoters, such as phytogenic feed additives, probiotics, prebiotics, synbiotics, and citric acids, have been explored for their efficacy in promoting growth and maintaining health without compromising food safety [3,6,7]. Citric acid improves nutrient digestion, regulates intestinal pH, and reduces harmful bacteria like *E. coli* [8,9] while promoting beneficial organisms like *Lactobacillus*, *Bifidobacterium*, and *Enterococcus* species. Probiotics balance gut microbiota, enhance digestion, and strengthen immunity [10]. Synbiotics, combining probiotics and prebiotics, improve feed efficiency, boost immunity, and increase disease resistance [11]. These additives collectively enhance gut health, growth performance, and sustainability in poultry farming. Small- to mid-scale farmers in Bangladesh are faced with poverty [12] and illiteracy, which limits their access to proper equipment for efficiently mixing feed additives. As a result, adding these additives to the water rather than the feed could provide a more practical and effective solution [13] for farmers in developing countries like Bangladesh. Based on previous studies on poultry with varying doses of these additives, we hypothesized that the inclusion of citric acids, synbiotics, and probiotics in drinking water could improve broiler performance while positively influencing blood profile parameters. The objective of this study was to evaluate the impact of citric acids, synbiotics, and probiotics on growth performance, meat yield, dressing parameters, and the blood profile of Cobb-500 broiler chickens.

## 2. Materials and Methods

This experiment was conducted at the experimental poultry farm of Hajee Mohammad Danesh Science and Technology University (HSTU), Dinajpur, Bangladesh. All the experimental procedures involving animals were revised and approved by the Animal Ethics Committee of Hajee Mohammad Danesh Science and Technology University.

### 2.1. Additives Procurement and Composition

Citric acid was purchased from Merck Specialties Private Limited, (Warli, Mumbai, India). Synbiotics were sourced from K.M.P. Biotech Co., Ltd., (Chonburi, Thailand) with each gram containing *Bacillus subtilis* (≥5 × 10^9^ CFU/g), *Pediococcus acidilacticii* (≥5 × 10^9^ CFU/g), *Enterococcus faecium* (≥5 × 10^9^ CFU/g), *Saccharomyces cerevisiae* (≥1 × 10^9^ CFU/g), and Xylo-oligosaccharide (35 g). Probiotics were obtained from Sanzyme Biologics (P) Ltd. (Hyderabad, India), with each gram containing *Bacillus subtilis* (≥4.5 × 10^9^ CFU/g), *Bacillus coagulans* (≥4.5 × 10^9^ CFU/g), and *Saccharomyces boulardii* (≥4.5 × 10^9^ CFU/g).

### 2.2. Experimental Design and Diet

A total of 400 one-day-old Cobb 500 broiler chickens (48.04 ± 1.36) g were randomly assigned to one of 4 treatment groups, with 4 replicates per treatment (25 birds per pen). The study included 3 additive treatments—Citric acid (CA) (2.5 g/L), Symbiotic (SB) (0.2 g/L), and Probiotic (PB) (0.5 g/L)—along with a control (CON) group that received no additives, and each was administered through drinking water for 35 days. Drinking water was changed in every 6 h interval from the 6 a.m. to 12 p.m. period of the day. The birds were raised in a semi-controlled shed where shed temperature and humidity were checked and measured four times a day (8 a.m., 12 p.m., 4 p.m., and 8 p.m.) using an automated thermo-hygrometer. Each pen measured 30 square feet (10 feet × 3 feet). The experimental shed and required related equipment were cleaned and disinfected using a 1% TH4+ solution (0.1-L diluted solution per square foot) and a 0.5% TH4+ solution, respectively (Manufactured by Sogeval, France, and distributed by Century Agro Ltd., Dhaka, Bangladesh). Fresh rice husk was used as the litter material and litter depth was maintained at 5–6 cm. Ad libitum feed and water were provided throughout the experiment. A commercial-type broiler diet (Table 1) divided into two different phases containing the starter phase d (0–14), and the grower d (15–35) was used as the experimental diet.

### 2.3. Performance and Carcass Characteristics

The body weight of the broilers was determined on d 0, 7, 14, 21, and 35, and feed intake (FI) was assessed daily by accounting for the feed offered and the leftovers. The body weight gain (BWG) and feed conversion ratio (FCR) were calculated using recorded data at each wk. To assess dressing yield, 80 randomly selected birds (5 birds per replication) were sacrificed by the cervical dislocation method on d 35. The live weight, dressed weight, thigh weight, breast meat weight, drumstick weight, and wing meat weight of slaughtered birds were collected and weighed just after sacrifice by a trained professional. The organ index was calculated using the formula: organ weight/live body weight × 100.

### 2.4. Blood Biochemistry

Approximately 3–4 mL of blood was collected from each replicate bird into sterile glass test tubes. The tubes were positioned at a 45° angle at room temperature to allow for clotting. After 2 h, the separated serum was carefully transferred into Eppendorf tubes and centrifuged at 3000 rpm for 10 min. The supernatant serum was then transferred to new Eppendorf tubes, properly labeled with a permanent marker for easy identification during chemical analysis and stored at −20 °C until further analysis. The serum lipid profile, including cholesterol, triglycerides, high-density lipoprotein (HDL), and low-density lipoprotein (LDL), was determined following standard procedures. Briefly, total cholesterol and triglycerides were determined using spectrophotometry (Spectronic, Genesis 5, Melville, NY, USA) following the methods described by [14]. HDL was analyzed by incubating 200 μL of serum with 500 μL of reagent at room temperature for 10 min before measuring absorbance at 505 nm. The concentrations (expressed in mg/dL) of cholesterol, triglycerides, HDL, and LDL were calculated using the following formulas.Cholesterol (mg/dL) = (Absorbance of sample/Absorbance of standard) × 200Triglycerides (mg/dL) = (Absorbance of sample/Absorbance of Standard) × Concentration of standardHDL (mg/dL) = (A_505nm_) × 320; A 505 nm (Sample read) is the absorbance value of the sample measured at 505 nm wavelength.LDL (mg/dL) = (Total serum cholesterol-Triglyceride/5) × HDL

### 2.5. Statistical Analysis

Data were analyzed using the analysis of variance techniques to determine significant differences between the means of different treatment groups in SAS 9.4 software (SAS Institute Inc., Cary, NC, USA, 2000), following the principles of a completely randomized design. To determine significant differences among treatments, the Duncan Multiple Range Test was performed. Statistical significance was set at *p* < 0.05, while trends were between 0.05 to 0.1.

## 3. Results

### 3.1. Broiler Performance

The effects of citric acid, synbiotic, and probiotic-treated drinking water on broiler growth performance, FI, and FCR are summarized in Table 2. Feed intake showed significant variation (*p* < 0.05) in the 5th week and overall, with the CON group consuming the highest amount of feed (2984.18 g), followed by the CA, SB, and PB groups. Significant differences (*p* < 0.05) were observed in BW and BWG from the 4th week onward. In the 4th week, the probiotic group demonstrated the highest BW (1533.52 g) and BWG (463.84 g) compared to the CON group (1482.76 g and 258.41 g, respectively). In the 5th week, the CA group exhibited a higher BWG (274.59 g) than the CON group (258.41 g). By the end of the trial, the PB group achieved the highest final BW (1746.66 g) and BWG among all dietary groups. FCR also differed significantly (*p* < 0.05), with the PB group showing the best FCR (1.62) compared to the CON group (1.74) during the 4th and 5th weeks, and the entire experimental period. No significant differences in FCR were observed between the CA (1.66) and SB (1.65) groups compared to the CON group.

### 3.2. Carcass Yield

Table 3 reveals that there were no significant differences in meat yield and bone development across the dietary treatment groups. However, the CA, SB, and PB groups showed numerically higher dressing percentages than the CON group. According to Table 4, there were no significant differences in head, neck, heart, and leg weight relative to body weight among the dietary groups.

### 3.3. Blood Biochemistry

In Table 4, the CON group had significantly the highest cholesterol and LDL levels (*p* < 0.05) compared to the other treated groups. No significant differences were observed in triglycerides and HDL levels among the treatment groups. All the supplemented groups displayed lower total cholesterol (*p* = 0.013) and LDL (*p* = 0.039) levels compared to the CON group.

## 4. Discussion

The results from the present study demonstrate that broilers supplemented with probiotic, synbiotic, and citric acid additives exhibited significant improvements in growth performance, FI, and FCR compared to the CON group. Probiotic supplementation showed the most consistent positive effects on BW and BWG, particularly from the 4th week onwards, aligning with studies that have highlighted probiotics role in improving feed utilization and enhancing growth through the modulation of gut microbiota [15]. This is supported by the findings of Rehman et al. and Biswas et al. [16,17], where probiotics supplemented through diet-enhanced BW and feed conversion efficiency in broilers by promoting more efficient nutrient absorption and better digestive health. In the present study, we observed a decrease in feed intake in treatment groups compared to the control. However, this finding aligns with the concept that feed intake is not always directly correlated with growth performance [18]. Several studies have mentioned increased feed intake with probiotic supplementation [19], while others have observed no effect [20], making efficiency of feed utilization the key factor. Probiotics may enhance the birds’ ability to digest and absorb nutrients, leading to better nutrient utilization [21,22] and reduced feed consumption.

The higher growth rates in the synbiotic-supplemented group are consistent with [23], who have shown that the synergistic effects of prebiotics and probiotics in synbiotics improve gut health and nutrient absorption, leading to better growth outcomes. Furthermore, synbiotics have been reported to optimize feed utilization without necessarily increasing FI [24], which aligns with the results observed in this study where synbiotic-treated broilers showed a lower FI than the CON but a better FCR. In contrast, several studies have reported no significant effects of synbiotic supplementation on the growth performance of broiler chickens [25,26,27]. These inconsistencies in results may be due to variations in factors such as the genetic background of the birds, differences in probiotic strains and dosages, and the type and inclusion level of prebiotics used [28].

While not as impactful as probiotics, citric acid supplementation also improved growth performance. The increase in BW and BWG observed with citric acid agrees with studies that suggest organic acids can lower gastrointestinal tract pH, inhibit pathogenic bacteria, and improve nutrient digestibility [29,30]. The significant improvement in FI and FCR in the CA group compared to the CON group supports its beneficial effects on feed consumption and feed efficiency [9]. Similar effect on gosling was reported with citric acid supplementation [31]. In contrast, Ref. [32] reported that high doses of CA (40 and 60 g/kg diet) suppressed growth performance in broiler chickens, attributing these adverse effects to nutrient and mineral metabolism disruption. The improvements in FCR observed in both SB and PB groups are consistent with previous findings [33,34], where PB and SB treatments led to improved feed conversion. The synergistic effects of probiotics and prebiotics in synbiotics appear to enhance feed utilization efficiently, as observed in this study.

The results of the present study indicated no significant effects of CA, SB, and PB supplementation on broiler dressing percentage, meat yield (breast, thigh, and wing meat), or bone development (thigh, drumstick, and wing bones) compared to the control group. These findings are consistent with the observations of [35], who also reported no significant impact of probiotics on carcass yield. While the additive treatments in the current study may have primarily enhanced gut health and fat metabolism, they may not have significantly promoted protein synthesis or muscle growth, resulting in minimal effects on carcass yield [36]. The differences in results may be attributed to variations in the method of probiotic administration [37]; our study used drinking water, whereas others supplemented through the diet. Broilers in the CON group had a higher percentage of abdominal fat than those in the treated groups. This is consistent with the findings of Islam et al., and Elbaz et al. [38,39], although their studies did not report differences in carcass percentage between CON and PB groups.

A significant difference was observed in blood cholesterol and LDL levels among the different treatment groups when various additives were supplied in broiler drinking water. The CA, SB, and PB groups exhibited significantly lower cholesterol and LDL levels than the CON group. This finding is in line with previous studies, such as [40], who reported that probiotics administered via drinking water reduced plasma cholesterol and triglyceride concentrations. Similarly, Ref. [41] found that the supplementation of *Lactobacillus sporogenes* lowered serum cholesterol and LDL levels. CA supplementation led to significantly lower cholesterol and LDL levels compared to the CON group, aligning with previous reports [42]. However, opposite findings were reported by [43], who observed no significant effects of citric acid on cholesterol and LDL levels. While there is limited research on the effects of feed additives on serum biochemical parameters when delivered via drinking water, these results provide insights into the complex interactions between additives and broiler metabolism. Further studies are needed to clarify the effects of these additives on lipid profiles when administered through drinking water.

Although the tested feed additives demonstrated improved results compared to the control (CON) diet, their underlying mechanisms of action were not investigated in this study. Future research focusing on intestinal microbial populations and the digestibility of both macro- and micronutrients may provide valuable insights into how these additives enhance broiler performance.

## 5. Conclusions

In conclusion, all three supplementations in drinking water improved broiler growth performance, with increased BWG and reduced FCR, where the highest BW, BWG, and best FCR were observed in the PB group. No significant differences were found in meat yield or organ weights, but all three groups showed better lipid profiles compared to the CON group. These results suggest that probiotics can enhance growth and feed efficiency the most in broilers, warranting further exploration of their metabolic effects.

## Figures and Tables

**Table 1 animals-15-01168-t001:** Ingredient and nutrient composition of experimental diets (“as is” basis) ^1^.

Item	Starter (0–14 Days)	Grower (15–35 Days)
Ingredients (%)		
Maize	50.32	53
Rice polish	8.00	10.0
Soybean	29.00	22.50
Protein concentrates (CP, 60%)	8.00	8.00
Oyster shell	1.00	1.00
Salt	0.30	0.25
DL-Methionine (99%)	0.20	0.18
Lysine (98.5%)	0.03	0.03
Vitamin-mineral premix ^2^	0.25	0.25
Soybean oil	3.50	4.00
Di calcium phosphate	0.25	0.25
Choline chloride (50%)	0.10	0.10
**Total**	100	100
Analyzed Value		
ME Kcal/kg	3050	3150
CP%	23.66	20.59
DM%	89.68	87.09
Lys%	1.05	1.06
Met%	0.65	0.63
Met + Cys%	1.00	0.92
Ca%	0.95	1.05
Available P%	0.45	0.42

^1^ Provided per kg of complete diet: 37.5 mg Zn (as ZnSO_4_); 37.5 mg Mn (as MnO_2_); 37.5 mg Fe (as FeSO_4_•7H_2_O); 3.75 mg Cu (as CuSO_4_•5H_2_O); 0.83 mg I (as KI); and 0.23 mg Se (as Na_2_SeO_3_•5H_2_O). ^2^ Provided per kg of complete diet: 12,000 IU of vitamin A, 4000 IU of vitamin D3, 37.5 IU of vitamin E, 4.5 mg of vitamin K3, 2.2 mg of Thiamin, 1.2 mg of Rivoflavin, 3 mg of vitamin B6, 0.03 mg of vitamin B12, 50 mg of Niacin, 0.5 mg of Folic acid, 0.08 mg of Biotin and 13.5 mg of Ca-Pantothenate.

**Table 2 animals-15-01168-t002:** Growth performance of broiler in different dietary treatments at different ages ^1^.

Parameter	Treatments ^2^				*p*-Value
	CON	CA	SB	PB	
Feed Intake					
Week 1	167.50 ± 1.33	170.57 ± 4.51	166.65 ± 3.47	162.06 ± 5.25	0.856
Week 2	537.34 ± 4.71	535.11 ± 4.29	535.94 ± 2.94	534.12 ± 1.77	0.932
Week 3	767.93 ± 2.45	770.44 ± 0.66	770.14 ± 0.93	771.50 ± 0.36	0.366
Week 4	829.70 ± 1.29	826.52 ± 1.53	828.35 ± 0.41	827.78 ± 0.61	0.279
Week 5	681.71 ^a^ ± 3.41	631.80 ^b^ ± 4.74	619.13 ^bc^ ± 4.59	609.29 ^c^ ± 6.62	0.00
Overall	2984.18 ^a^ ± 12.4	2934.44 ^c^ ± 10.09	2920.2 ^b^ ± 8.78	2904.75 ^bc^ ± 14.56	0.03
Body weight, g					
Initial	45.7 ± 0.03	44.3 ± 0.08	44.6 ± 0.02	45.5 ± 0.05	0.923
Week 1	206.78 ± 2.15	211.53 ± 1.92	209.58 ± 2.83	212.57 ± 1.16	0.294
Week 2	541.15 ± 6.09	548.67 ± 6.98	548.67 ± 4.98	551.23 ± 2.68	0.607
Week 3	1062.46 ± 4.41	1051.47 ± 9.79	1047.32 ± 3.73	1069.68 ± 4.80	0.377
Week 4	1482.76 ^c^ ± 1.63	1496.65 ^bc^ ± 5.15	1503.86 ^b^ ± 8.48	1533.52 ^a^ ± 5.66	0.002
Week 5	1741.17 ^c^ ± 0.98	1771 ^ab^ ± 11.51	1772.68 ^b^ ± 10.90	1792.16 ^a^ ± 5.89	0.018
Body weight gain, g					
Week 1	161.06 ± 2.56	167.23 ± 4.38	164.98 ± 2.78	167.07 ± 3.69	0.726
Week 2	334.37 ± 4.05	337.14 ± 5.06	339.09 ± 5.22	338.66 ± 2.43	0.864
Week 3	521.31 ± 8.97	502.80 ± 16.1	498.65 ± 1.29	518.45 ± 7.46	0.542
Week 4	420.30 ^c^ ± 3.18	445.18 ^b^ ± 12.18	456.54 ^b^ ± 4.76	463.84 ^a^ ± 8.24	0.021
Week 5	258.41 ^c^ ± 2.44	274.59 ^a^ ± 6.52	268 ^b^ ± 2.53	258.64 ^ab^ ± 3.39	0.047
Overall	1695.47 ^c^ ± 8.66	1726.70 ^b^ ± 5.23	1728.08 ^ab^ ± 10.78	1746.66 ^a^ ± 7.64	0.027
Feed conversion ratio					
Week 1	1.04 ± 0.02	1.02 ± 0.02	1.01 ± 0.01	0.97 ± 0.02	0.873
Week 2	1.61 ± 0.00	1.59 ± 0.01	1.58 ± 0.02	1.58 ± 0.02	0.481
Week 3	1.47 ± 0.02	1.54 ± 0.07	1.54 ± 0.00	1.49 ± 0.04	0.506
Week 4	1.97 ^a^ ± 0.01	1.85 ^bc^ ± 0.05	1.81 ^b^ ± 0.02	1.78 ^c^ ± 0.06	0.012
Week 5	2.64 ^a^ ± 0.02	2.32 ^ab^ ± 0.06	2.30 ^b^ ± 0.03	2.25 ^c^ ± 0.02	0.508
Overall	1.74 ^a^ ± 0.03	1.66 ^b^ ± 0.08	1.65 ^b^ ± 0.04	1.62 ^c^ ± 0.09	0.042

^1^ Abbreviation: CON, Control; CA, Citric acid; SB, Synbiotic; PB, Probiotic. ^2^ Data are presented as mean ± SE. Data represent 25 birds/pen and 4 pens/treatment. ^a,b,c^ indicate differences between groups. *p* < 0.05 was considered as significant.

**Table 3 animals-15-01168-t003:** Meat yield and bone development of broiler in different dietary treatments ^1^.

Parameter	Treatments ^2^				*p*-Value
	CON	CA	SB	PB	
Meat yield (%)					
Dressing Percentage (%)	58.37 ± 2.24	57.10 ± 1.32	56.82 ± 0.78	58.41 ± 0.47	0.780
Breast meat	20.12 ± 0.84	17.89 ± 0.35	18.36 ± 0.68	20.15 ± 0.80	0.103
Thigh meat	8.12 ± 0.13	7.77 ± 0.12	7.51 ± 0.14	8.07 ± 0.51	0.117
Drumstick meat	6.39 ± 0.24	5.54 ± 0.19	5.30 ± 0.08	6.02 ± 0.40	0.062
Wing meat	3.44 ± 0.04	3.27 ± 0.05	3.32 ± 0.05	3.39 ± 0.03	0.058
Bone development (%)					
Thigh bone	2.01 ± 0.15	2.04 ± 0.05	1.97 ± 0.04	1.93 ± 0.18	0.911
Drumstick bone	2.84 ± 0.06	2.72 ± 0.07	2.88 ± 0.17	2.84 ± 0.04	0.691
Wing bone	2.84 ± 0.06	2.91 ± 0.00	2.81 ± 0.09	3.07 ± 0.29	0.066

^1^ Abbreviation: CON, Control; CA, Citric acid; SB, Synbiotic; PB, Probiotic. ^2^ Data are presented as mean ± SE. Data represent 5 birds/replication pen. Total 20 birds/treatment.

**Table 4 animals-15-01168-t004:** Blood biochemical parameters of broiler in different dietary treatments (mg/dL) ^1^.

Parameter	Treatments ^2^				*p*-Value
	CON	CA	SB	PB	
TC	138.83 ^a^ ± 11.89	108.96 ^b^ ± 1.53	101.29 ^c^ ± 18.61	112.62 ^b^ ± 5.94	0.013
Triglycerides	82.03 ± 5.42	82.03 ± 5.42	67.34 ± 4.00	65.30 ± 2.35	0.090
HDL	37.06 ± 1.10	37.76 ± 2.06	37.28 ± 0.14	33.48 ± 0.14	0.107
LDL	88.93 ^a^ ± 10.94	56.08 ^bc^ ± 0.22	50.06 ^c^ ± 17.63	62.62 ^b^ ± 6.58	0.039

^1^ Abbreviation: CON, Control; CA, Citric acid; SB, Synbiotic; PB, Probiotic; HDL, High-Density Lipoprotein; LDL, Low-Density Lipoprotein; TC, Total Cholesterol. ^2^ Data are presented as mean ± SE. Data represent 5 birds/replication pen. Total 20 birds/treatment. ^a,b,c^ indicate differences between groups. *p* < 0.05 was considered as significant.

## Data Availability

The original contributions presented in this study are included in the Appendix A. Further inquiries can be directed to the corresponding author.

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
