# Peer review of "Effects of Citric Acid, Synbiotic, and Probiotic Supplementation Through Drinking Water on Growth Performance, Carcass Yield, and Blood Biochemistry of Broiler Chickens"

_animals, 2025, doi:10.3390/ani15081168_

Round 1
Reviewer 1 Report
Comments and Suggestions for Authors
Antimicrobial reduction and use of gut health additives is an important area and thus work like this is important. I have a few suggestions to increase the impact of this manuscript.
Section 2.1: Were the probiotic and synbiotic products used designed for water application? I ask this as many Bacillus based products are designed for in feed application and so they don't disolve well in water which can impact efficacy when administered in water.
Lines 117-119: How many birds were weighed to get body weights?
Table 3: What is DP(%)?
Table 4: The title "Dressing parameters of broiler in different dietary" should this be "Dressing parameters of broiler in different dietary treatments"?
Lines 189-190: "In Table 5, the CON group had significantly the highest cholesterol and low-density lipoprotein (LDL) levels (P<0.05) compared to the CON and other treated groups" Consider rewording as it doesnt make sense to compare the CON group with itself?
Lines 211-213: "...making efficiency of feed utilization the key factor." Perhaps the authors could show some references where probiotic/synbiotics/organic acids have improved feed efficiency/FCR to support this statement.
Lines 234-235: "The synergistic effects of probiotics and prebiotics in synbiotics appear to enhance feed utilization more efficiently than either additive alone, as observed in this study." You can not conclude this from this study. The bacteria in the probiotic and the synbiotic are not the same thus you don't know whether the inclusion of the prebiotic in the synbiotic is making a difference. The difference between the SB and PB groups could be due to the different bacteria present. Also there is not a "prebiotic only" group.
Lines 245-248: "Both CA and PB significantly increased gizzard weight compared to the CON, likely due to the acidifying effect of citric acid, which lowered gut pH and enhanced gizzard activity by probiotics improved nutrient digestion, ultimately promoting gizzard development
[35,36]." This sentence does not really make sense and should be rewritten. The PB group do not have citric acid therefore you can't say that the increased gizzard weight in this group is due to the acidifying effect of the citric acid. Furthermore, gizzard development is more driven by feed texture and fibre inclusion rather than pH. Additionally for references 35 and 36, whilst they do talk about gizzard develeopment, they don't really discuss gizzard development in terms of treatments such as those used in this study. These references are not appropriate for this statement.
Lines 253-254: "Broilers in the CON group had a higher percentage of abdominal fat than those in the treated groups." Why might this be?
Other comments:
- The abstract discusses these additives as an alternative to antibiotic growth promotors however in the CON diet does not contain any AGPs. As such this experiment does not assess the products as alternatives to AGPs.. it seems like a missed opportunity when the abstract discusses antibiotics.
- Do you have mortality figures for the trial it would be interesting to see whether there was an impact.
- Rather than just putting a significant effect for the treatment group perhaps you could specify where the significant differences are between treatments/control. Linking with this, perhaps the authors could then suggest which additive treatment is better using the significant differences for each measurement.
- In some of the groups the SE is much greater for some of the measurements e.g. Table 2 week 3 body weight CON 1062.46±4.41, CA 1051.47±17.24, SB 1047.32±3.73, PB 1069.68±4.80. The SE for CA is much greater than the other groups which would imply there was something different about the birds in that group. Do the authors know why there was such a big difference?
- In the discussion the authors mention a couple of examples where probiotic bacterial strains have conferred certain health benefits. Perhaps they could include examples from the species present in the products used?
Author Response
Comment 1: Section 2.1: Were the probiotic and synbiotic products used designed for water application? I ask this as many Bacillus based products are designed for in feed application and so they don't disolve well in water which can impact efficacy when administered in water.
Comment 1: Author response: yes, the products were purchased for their suitability to mix well with water. Their doses were also recommended by the producer companies.
Comment 2: Lines 117-119: How many birds were weighed to get body weights?
Comment 2: Author response: all the birds were weighed to get body weights. They were counted as pen not individual.
Comment 3: Table 3: What is DP(%)?
Comment 3: Author response: it is dressing percentage. We have written it in full now at table 3.
Comment 4: Table 4: The title "Dressing parameters of broiler in different dietary" should this be "Dressing parameters of broiler in different dietary treatments"?
Comment 4: Author response: We have already discarded this table as another reviewer suggested it to be non-relevant.
Comment 5: Lines 189-190: "In Table 5, the CON group had significantly the highest cholesterol and low-density lipoprotein (LDL) levels (P<0.05) compared to the CON and other treated groups" Consider rewording as it doesnt make sense to compare the CON group with itself?
Comment 5: Author response: We have corrected it in line 184, page 5.
Comment 6: Lines 211-213: "...making efficiency of feed utilization the key factor." Perhaps the authors could show some references where probiotic/synbiotics/organic acids have improved feed efficiency/FCR to support this statement.
Comment 6: Author response: This discussion part was about probiotic. So, the authors think adding citations for probiotic, symbiotic and organic acid is not necessary here. We have added few more related references to related discussion sections. Probiotic (line 209) & citric acid (Line 226).
Comment 7: Lines 234-235: "The synergistic effects of probiotics and prebiotics in synbiotic appear to enhance feed utilization more efficiently than either additive alone, as observed in this study." You cannot conclude this from this study. The bacteria in the probiotic and the synbiotic are not the same thus you don't know whether the inclusion of the prebiotic in the synbiotic is making a difference. The difference between the SB and PB groups could be due to the different bacteria present. Also, there is not a "prebiotic only" group.
Comment 7: Author response: Yes, thank you for indicating it. We have re-worded the sentence to remove the wrong conclusion.
Comment 8: Lines 245-248: "Both CA and PB significantly increased gizzard weight compared to the CON, likely due to the acidifying effect of citric acid, which lowered gut pH and enhanced gizzard activity by probiotics improved nutrient digestion, ultimately promoting gizzard development
[35,36]." This sentence does not really make sense and should be rewritten. The PB group do not have citric acid therefore you can't say that the increased gizzard weight in this group is due to the acidifying effect of the citric acid. Furthermore, gizzard development is more driven by feed texture and fiber inclusion rather than pH. Additionally for references 35 and 36, whilst they do talk about gizzard development, they don't really discuss gizzard development in terms of treatments such as those used in this study. These references are not appropriate for this statement.
Comment 8: Author response: Actually, another reviewer indicates this part to be not related to the objective of this study. We also think that it is not relevant to our hypothesis, so we have deleted this part’s M&M, results, discussion and references.
Comment 9: Lines 253-254: "Broilers in the CON group had a higher percentage of abdominal fat than those in the treated groups." Why might this be?
Comment 9: Author response: : Actually, another reviewer indicates this part to be not related to the objective of this study. We also think that it is not relevant to our hypothesis, so we have deleted this part’s M&M, results, discussion and references.
Other comments:
Comment 10: The abstract discusses these additives as an alternative to antibiotic growth promotors however in the CON diet does not contain any AGPs. As such this experiment does not assess the products as alternatives to AGPs.. it seems like a missed opportunity when the abstract discusses antibiotics.
Comment 10: Author response: well, it was really a short coming of this study. That’s why we have written it as non-antibotic additives. We could not focus more o the antiotic alternatives. In future, we will keep this suggestion in mind to consider a positive control group.
Comment 11: Do you have mortality figures for the trial it would be interesting to see whether there was an impact.
Comment 11: author response: We do not have a specific mortality calculation for this study. Mortality in each group did not exceed 6%. Therefore, we did not consider putting it in statistical analysis.
Comment 12: Rather than just putting a significant effect for the treatment group perhaps you could specify where the significant differences are between treatments/control. Linking with this, perhaps the authors could then suggest which additive treatment is better using the significant differences for each measurement.
Comment 12: Author response: We planned this study for finding better feed additives easy to use by the small scale farmers. We have conducted Duncan test that indicated probiotic being the best in case of BW, BWG, and FCR. A preplanned contrast would have been better. However, in our next experiment focusing only on the probiotic, we will consider dose effect and positive control to indicate specific dose and effect difference.
Comment 13: In some of the groups the SE is much greater for some of the measurements e.g. Table 2 week 3 body weight CON 1062.46±4.41, CA 1051.47±17.24, SB 1047.32±3.73, PB 1069.68±4.80. The SE for CA is much greater than the other groups which would imply there was something different about the birds in that group. Do the authors know why there was such a big difference?
Comment 13: Author response: There was some calculation error on CA group. We have recalculated from the raw data and the corrections were done in Table 2.
Comment 14: In the discussion the authors mention a couple of examples where probiotic bacterial strains have conferred certain health benefits. Perhaps they could include examples from the species present in the products used?
Comment 14: Author response: getting previous studies with similar species combination is difficult. We have tried to add new references for probiotic (Line 209) and citric acid (line 226). However, symbiotic with the same species combination are scares.
Reviewer 2 Report
Comments and Suggestions for Authors
Major Comments
- The reduced liver size, lower lipid profile, and decreased abdominal fat weight are intriguing. Future studies could explore hepatic lipid metabolism, lipolysis, or the gut microbiome’s role in the Gut-Liver Axis. However, if the data and discussion needed to interpret these interesting points are insufficient, and the changes in carcass performance are not used to interpret these lipid changes, or vice versa, it may be worth reconsidering whether this data is essential for the paper.
Minor Comments
- What are the economic costs of these additives? Are they affordable for farmers, or do they pose a financial burden? Was the dilution concentration of the additives determined with economic feasibility in mind?
- Was the serum lipid profile analysis conducted as an indicator of animal health, or was it measured with consideration for human consumption? Please clarify the rationale.
- Please spell out abbreviations in full when they first appear, except in the abstract.
- Have t-test results been considered for dressing parameters and blood biochemical parameters in comparison to the control, rather than relying solely on ANOVA?
Author Response
Major Comments
Comment 1: The reduced liver size, lower lipid profile, and decreased abdominal fat weight are intriguing. Future studies could explore hepatic lipid metabolism, lipolysis, or the gut microbiome’s role in the Gut-Liver Axis. However, if the data and discussion needed to interpret these interesting points are insufficient, and the changes in carcass performance are not used to interpret these lipid changes, or vice versa, it may be worth reconsidering whether this data is essential for the paper.
Comment 1: Author response: After considering, the authors also think this part is not related to the objective of this study. Therefore, we have deleted table 4 and related abstract (line 37), materials and methods (Line 123), results (Line 178) and discussion (Line 249) from the manuscript. References (35-41) were also rearranged for the changes.
Minor Comments
Comment 2: What are the economic costs of these additives? Are they affordable for farmers, or do they pose a financial burden? Was the dilution concentration of the additives determined with economic feasibility in mind?
Comment 2: Author response: Local farmers in Bangladesh are using Antibiotic as a growth promoter, also as a preventive dose in every week. Which is much more costly compared to the feed additives tested here. The doses were used as the doses recommended by the feed additive producers. They are feasible compared to the cost of using antibiotics. Though the result is not as like as antibiotic, these feed additives will provide option for antibiotic alternative and will meet consumer preferences.
Comment 3: Was the serum lipid profile analysis conducted as an indicator of animal health, or was it measured with consideration for human consumption? Please clarify the rationale.
Comment 3: Author response: Yes, they were measured as an indicator of health of broilers.
Comment 4: Please spell out abbreviations in full when they first appear, except in the abstract.
Comment 4: Author Response: All the abbreviations were checked and spelled out.
Comment 5: Have t-test results been considered for dressing parameters and blood biochemical parameters in comparison to the control, rather than relying solely on ANOVA?
Comment 5: Author response: Rather than t-test, we have conducted Duncan test to identify difference between groups. We think it should be enough to identify differences.
Reviewer 3 Report
Comments and Suggestions for Authors
The present study investigated the effects of citric acid, symbiotic, and probiotic supplementation on the growth performance, carcass yield, and blood biochemistry of broiler chickens. Following are some comments on the manuscript. L63-65: Existing references have shown the effects of citric acid on numerous growth parameters of broiler chickens. What is the superiority of your research compared to other researches? L96: The doses presented here are different with the abstract. Which is correct? The criteria for these doses should be clearly provided in the main text. L97: The additives were supplemented into the water; more details should be provided. E.g. What was the change frequency of water? Were these additives stable in the water? L146: Did the authors test the normal distribution of data? Table 5: What is TRT4 here? Tables: The sampling number should be clearly shown in all table legends. Discussion: This study is only a description study; no mechanism was involved. The authors should propose the future work to illustrate the potential mechanism of these additives on the growth of broilers in the discussion part.
Author Response
Comment 1: L63-65: Existing references have shown the effects of citric acid on numerous growth parameters of broiler chickens. What is the superiority of your research compared to other researches?
Author response: The superiority of your research is that it was conducted in a Bangladeshi environment with farmers rearing condition. Many of the beneficial feed additives cannot be used in Farmer’s level in Bangladesh and their effectivity might be reduced due to the farm facility and environmental condition.
Comment 2: L96: The doses presented here are different with the abstract. Which is correct? The criteria for these doses should be clearly provided in the main text.
Author response: The abstract doses are the correct one. So, changes were made in Line 95, page 2.
Comment 3: L97: The additives were supplemented into the water; more details should be provided. E.g. What was the change frequency of water? Were these additives stable in the water?
Author response: The water was changed every 6 hours of a day from morning 6 Am to night 12 PM. It has been changed in line 97- 98 page 3. The additives were water soluble and commercially available for water mixing. Therefore, we did not check their stability in water.
Comment 4: L146: Did the authors test the normal distribution of data?
Author response: Yes, Normality of data distribution was checked before statistical analysis. It was within range.
Comment 5: Table 5: What is TRT4 here?
Author response: It was a writing mistake. It has been deleted from line 189.
Comment 6: Tables: The sampling number should be clearly shown in all table legends.
Author response: The sample number per treatment has been added at the legend of each table.
Comment 7: Discussion: This study is only a description study; no mechanism was involved. The authors should propose the future work to illustrate the potential mechanism of these additives on the growth of broilers in the discussion part.
Author response: We have added a part at the discussion section explaining this at Line 259-263, page 7.
Round 2
Reviewer 1 Report
Comments and Suggestions for Authors
Happy with the edits.